# How Collective Intelligence Fosters Incremental Innovation

**Jung-Yong Lee [1] and Chang-Hyun Jin [2,*]** 

[1]    Department of Business Administration, SungKyunKwan University, Seoul 03063, Korea
[2]    Department of Business Administration, Kyonggi University, Suwon 16227, Korea
*    Correspondence: chjin@kgu.ac.kr

**Abstract:** The study aims to identify motivational factors that lead to collective intelligence and understand how these factors relate to each other and to innovation in enterprises. The study used the convenience sampling of corporate employees who use collective intelligence from corporate panel members ($n = 1500$). Collective intelligence was found to affect work process, operations, and service innovation. When corporate employees work in an environment where collective intelligence (CI) is highly developed, work procedures or efficiency may differ depending on the onset of CI. This raises the importance of CI within an organization and implies the importance of finding means to vitalize CI. This study provides significant implications for corporations utilizing collective intelligence services, such as online communities. Firstly, such corporations vitalize their services by raising the quality of information and knowledge shared in their workplaces; and secondly, contribution motivations that consider the characteristics of knowledge and information contributors require further development.

**Keywords:** collective intelligence; social contribution motivation; personal contribution motivation; work process; operation; service innovation; incremental innovation

---

## 1. Introduction

The ubiquity of information communication technologies—increasingly brought on by recent breakthrough developments—is apparent as a means of inducing intrinsic yet innovative change across corporate management practices. Competition is becoming among corporations that fight for survival amidst a rapidly changing global economic environment. As a result, corporations are keen on finding ways to build innovative capacity, which requires the introduction of collective intelligence as a means of solving problems through exchange and cooperation, with groups that are both internal and external to them.

Several researchers have raised the idea of launching a community focused on collective intelligence (hereafter CI) for the common good over recent years [1]. The challenges and opportunities in applying CI in enterprises are necessary to improve the effectiveness of decision-making or to solve various business issues [1,2]. The possibility of combining CI with existing corporate perspectives, such as rapid decision-making and efficient use of internal resources, has become a key factor in determining future sustainable development.

> *"Collective intelligence increases the capacity for effective action in pursuit of common aims and finding emergent and sustainable solutions to the complex problems." [3]*

The internal capacities of a corporation have perhaps reached its limit, and a new source of competitiveness is desperately needed to develop sustainable management practices. However, research on identifying cause-and-effect relationships associated with CI is lacking. Moreover, previous studies

do not build a theoretical model that explains the relationship between CI and incremental innovation. Therefore, this study aims to explore motivational factors that lead to CI and understand how these factors relate to each other and to innovation in enterprises. The results will provide valuable data to support effective decision-making and various organizational strategies.

## 2. Literature Review

### 2.1. Collective Intelligence (CI)

Research on CI has explored the capabilities of a collective [4–6]. CI as a topic has typically been studied and applied in the fields of sociology, business management, and computer science; however, today it is being applied across all social phenomena. CI, also known as swarm intelligence, has been a major interdisciplinary topic of research in the fields of entomology, sociology, computer science, and others [7]. CI enables an idea to have a greater impact than any individual's idea on its own as a result of a process whereby the individual idea is combined with and evolves with others, creating a synergy effect. Wikipedia, a free encyclopedia created by the public where thousands of web users contribute written and edited articles based on their knowledge and information, was the first example of a website showcasing the existence of CI. Wikipedia is considered by some to be the most successful case of public CI in history.

The definition of CI varies. It has been defined as the ability of a group to arrive at a solution that is better than what any of the members has achieved individually; it has also been defined as groups of individuals doing things collectively that seem intelligent [8,9]. Some scholars have defined

> "CI as the capacity of human communities to evolve towards higher-order complexity and harmony through such innovation mechanisms as variation-feedback-selection, differentiation-integration-transformation, and competition" [10]

CI entails a process whereby individuals share, cooperate, and integrate their knowledge, information, and ideas, resulting in an intellectual capability that surpasses the intellectual capability of individuals; this process also results in the capabilities of a collective formed by the participation in and sharing of information and knowledge of many individuals [9]. Thus, CI characteristics include the creation of new knowledge through co-creation with others, as well as leading participation. CI is assumed to be a collection of rational judgments made by individuals and is considered to be a better source of judgment than a small group of specialized professionals or the combined capabilities of an individual.

CI is a form of intelligence that results from the mutual engagement between people, which is thought to result in better solutions to problems than would otherwise be possible, because of the synergistic effects that result from the collective handling of tasks and the aggregation of advice and criticism [11]. CI is especially suited to work that demands innovation or creativity [5]. The CI approach adds a new perspective to technology in supporting energy awareness and eco-friendly choices [12,13]. Surowiceki argued that CI can move markets and society and that a collective always produces wiser judgments than an individual [4]. CI tools are conceived for situations of uncertainty about the impact of actions; it can also be a valuable marketing tool [10].

Based on methods of mutual engagement and cooperation level, Dutton classified CI into three types: (1) The sharing type; (2) the contributing type; and (3) the co-creating type [14]. These three classifications can be distinguished by their methods of mutual engagement. Sharing CI is typified by activities, such as a simple release of information and individual and one-on-one forms of mutual engagement. The second aspect is building CI through sharing, which includes tools to seek solutions to their problems and share and acquire new knowledge or information. Contributing CI is typified by activities, such as dialog among participants and active responses regarding opinions, knowledge, and information. It reshapes those who contribute information to the collective group [14]. The active responses considered in this instance involve the evaluation of opinions, information, and knowledge of one's counterpart that serve as the basis for developing opinions, information, and knowledge.

From this perspective, CI-building through participants includes users' positive participation in space to develop their personal knowledge, career, and capability recognition. Co-creating CI is typified by various active methods of mutual engagement, such as updating, editing, and debating, which seek to fine-tune opinions that lead to the production of knowledge or information. This includes the development of users' operation or management knowledge or information through CI. Thus, three types of CI-building through participation, sharing, and co-creation are applied to this study.

The general idea behind the collaboration is the notion that it elicits a self-driven response among individuals vying for a common goal [15]. In the case of Wikipedia, this involves active corrections and editing activities [16]. Leadbeater explains CI on the web-based on Web 2.0 characteristics, such as participation, sharing, and openness. CI requires participation [17]. Individuals must be able to easily express their opinions, register new viewpoints, and solicit opinions from other sources. Thus, CI emerges through continual voluntary participation of individuals.

Leadbeater emphasizes the importance of collaborative creativity and mentions participation, cooperation, openness, and sharing as necessary factors for CI. He argues that collaborative creativity results in the active formulation of ideas through the process of sharing and opening up of ideas between individuals that are combined later [15]. Sulis perceives CI as a probability-based aggregation of multiple semi-independent participants who communicate their ideas and freely engage in discussions within their environments [17]. Corporations explore various means of applying CI to business practices and are especially keen on utilizing it to establish creative problem-solving methods [18].

### 2.2. Motivation Theory

Motivation theory has long been employed in academic areas, such as social science and management. This study explores the motivational factors that lead to CI by examining the Theory of Planned Behavior (TPB) and the Theory of Reasoned Action (TRA) proposed by Fishein and Ajzen [19]. These theories explain that the psychological mechanisms which were affecting attitudes lead to certain behaviors. The motivation behind knowledge sharing in online communities can be explained as an organized civil movement not influenced by any direct or clear perceptions of compensation carried out at the individuals' discretion [20].

Other scholars also attempt to explain user motivations for knowledge sharing by applying the expected compensation from economic exchange theory, expected association from social exchange theory, and expected contribution from social cognitive theory as variables that affect knowledge sharing [21]. The subjective norms and perceived behavioral controls that underlie knowledge sharing are closely related to knowledge donation and knowledge collection [22]. Liao, To, and Hsu attempt to explain the knowledge-sharing attitudes of knowledge users by dividing them into groups motivated by utilitarian or hedonic motivations [23]. Wasko and Faraj add external rewards—a type of individual motivator—as a variable to explain the knowledge-sharing process [24].

The dispositions of people who contribute to CI may not differ from those who do not. Nov provides a pioneering insight into this notion in his study, exploring motivational factors among Wikipedians [25]. He adds two categories of motivation, behind the behavior of volunteers, to six others, posited in earlier studies: Protective, values-based, career, social, understanding, and enhancement—namely fun and ideology. This makes a total of eight contribution motivations.

Through a series of surveys, Rafaeli et al. uncovered—in order of importance—cognitive needs (e.g., the desire for information acquisition or intellectual challenge), affective needs (e.g., the desire for emotional experiences), and integrative needs (e.g., the desire to share knowledge with others or contribute to others) as the contribution motivation behind Wikipedia. They propose exploring the relative importance of psychological, social, communal, economic, gratifying, and mutually engaging aspects among various motivational factors as additional topics of research related to motivation [26].

Based on previous discussions, this study attempts to explain the motivation behind the use of CI by dividing the motives into groups reflecting social or individual objectives. In other words, the contribution motivations behind CI are categorized into social contribution motivations and

individual contribution motivations. The former includes knowledge sharing, content correction, answering others' questions, recognition of capability and acquisition of greater reputation, and a preference for knowledge cooperation. Individual contribution motivations include acquisition of new knowledge, expectations of responses from others, tangible and intangible compensations, an exhibition of knowledge, and benefits to career development.

### 2.3. Incremental Innovation

What is incremental innovation? Some scholars have defined it as "not-radical" [27], which can be seen as an unsatisfying definition. Taruss, Boit, and Korir defined this as

*"incremental improvements to existing products, services, and organizational routines that can enhance performance, quality, and usefulness, and are vital to making more competitively advanced products and services." [28–31]*

Incremental innovation vies for improvements in the status quo and is continual while improving on existing assets. Previous research divides incremental innovation into three categories: Work process and procedure innovation, operational innovation, and service innovation [8,32]. First, the work process and procedure innovation are related to the recycling and partial substitution of items associated with the production or specifications of a product. For instance, in terms of the field of service, this implies innovation associated with the simplification of management processes, work processes, or existing internal regulations; reduction of costs; and enhanced work efficiency [33,34].

Second, operational innovation includes innovations associated with redesigning methods and work plans for manufacturing products, and changes in methods, numbers of operations, and services related to reliability and quality. Third, service innovation includes innovations associated with the integration of lower-tier systems aimed at reconfiguring production systems that provide flexibility in operations and facilities. In the service sector, this entails innovations associated with improved services, customer satisfaction, and rapid customer service response to complaints [34]. In light of the above, this study attempts to understand how CI is related to incremental innovation, such as work processes, operations, and services innovation.

## 3. Hypotheses

### 3.1. The Relationship between Two Contribution Motivations and CI

Nam, Ackerman, and Adamic include altruism and the ability to learn as individual motivators of knowledge sharing [35]. Wasko and Faraj include external rewards—a type of individual motivator—as a variable to explain the knowledge-sharing process [24]. The process of sharing knowledge produces joy as it entails the thought of helping others. According to social exchange theory, knowledge contribution motivation variables affect attitudes regarding knowledge sharing, which in turn affects the intent to share knowledge, and the intent to share knowledge is closely related to the act of sharing knowledge [21]. According to the expanded TPB, intent to share knowledge is affected by attitudes regarding knowledge sharing, subjective norms regarding knowledge sharing, technical norms regarding knowledge sharing, knowledge sharing itself, and limitations of knowledge sharing. Thereafter, the intent to share knowledge affects the act of sharing knowledge [36]. Each individual, as one adaptive agent, contributes knowledge, communicates with another, refers to another, votes for others, modifies other's knowledge, and accomplishes the emergence of mass intelligence [37]. The notion of value co-creation described by service-dominant logic seems to be parallel to that of CI. Some scholars have mentioned that the ability to collectively harness intelligence represents a means of co-creating value in various aspects of modern business [38]. As shown in the above discussion, the assumptions that can be applied to motivation research regarding CI include categories of motivations, such as protective, value, career, social, understanding, and enhancement, as well as contribution motivations, such as cognitive needs, affective needs, and integrative needs [26].

Thus, such individual motivators and social orientation relationships can be expected to affect the onset of CI. Therefore, this study intends to explore two contribution motivations for CI.

**Hypothesis 1 (H1).** *Social contribution motivations are positively related to sub-factors of CI-building through participation (H1-1), sharing (H1-2), and co-creation (H1-3).*

**Hypothesis 2 (H2).** *Personal contribution motivations are positively related to sub-factors of CI-building through participation (H2-1), sharing (H2-2), and co-creation (H2-3).*

Evaluations of CI may vary depending on the level of active evaluation of information or knowledge. The degree to which knowledge or information is produced is also expected to become an important factor in evaluating CI [14,15,18]. Boder predicts that CI will play an essential role in corporate knowledge management. He notes that CI will bring new knowledge and innovation that will resolve core problems faced by corporations, and mentions the need to construct CI that includes the expertise of individuals associated with specialized knowledge in particular fields within an organization, as well as cultural norms, informal networks, and strategic market-related knowledge [39].

Engel et al. note that CI is a critical factor to improve decision-making in groups and to produce higher group performance [40]. CI was then shown to predict a team's future performance on more complex tasks [41]. It can be assumed that CI is built upon some combinations of individual members' attributes, group structures, processes, and norms.

Innovation is in demand not only from corporations that are keen on adapting to globalization and rapidly changing domestic and international environments, but also from governments and all other sectors that strive to bring changes to organizations and processes for enhancing competitiveness and adaptability. In general, innovation entails incremental or radical change associated with objects, thoughts, and the status of progress or services [42]. As mentioned above, incremental innovation naturally occurs in the work environment or corporate work processes. Several studies clarify the relationship between a corporation's organizational structure and culture with incremental innovation. This study predicts that CI—a form of intelligence that encompasses knowledge and information produced through user participation, sharing, and collaboration, and is created to improve individual work processes and procedures, product development, idea development, operations, and services—is closely related to incremental innovation. Most corporations emphasize process innovations that reduce costs, assist in product development, and make organizational management more efficient for satisfying customer needs.

Some scholars observe that CI emerges from a combination of bottom-up and top-down processes within groups and predict future performance and learning in a wide range of environment [41]. Individual intelligence is recognized as a means of understanding job performance in an organization, as well as for understanding many aspects of group performance, such as service process, task, and innovation in the organization [41]. Apart from playing an important role in improving corporate work processes, management procedures, and customer satisfaction, CI-building is also expected to lead to the information entropy process at the organizational level. The relationship between CI and incremental innovation may produce different forms of disorders in the organizational system, based on information uncertainty. Thus, an entropy process emerges. Under these conditions, CI-building requires internal control elements to ensure that the organizational system evolves toward higher-order levels, in terms of the amount of information. Considering the scholarly definition of incremental innovation, CI will play a significant role in improving corporate work processes, management procedures, and customer satisfaction services, along with enhancing work efficiency associated with idea development, product development, and strategic thinking. From the perspective, it is expected that CI-building leads to incremental innovation. Therefore, our hypotheses are stated as follows:

**Hypothesis 3 (H3).** *CI-building through participation positively affects work process innovation (H3-1), operation innovation (H3-2), and service innovation (H3-3) as sub-factors of incremental innovation.*

**Hypothesis 4 (H4).** *CI-building through sharing positively affects work process innovation (H4-1), operation innovation (H4-2), and service innovation (H4-3) as sub-factors of incremental innovation.*

**Hypothesis 5 (H5).** *CI-building through co-creation positively affects work process innovation (H5-1), operation innovation (H5-2), and service innovation (H5-3) as sub-factors of incremental innovation.*

*3.2. Suggested Research Model*

To guide the analysis of the data collected for this study, the study devised the following suggested research model to illustrate the relationships between the two contribution motivations, collective intelligence, and incremental innovation (Figure 1).

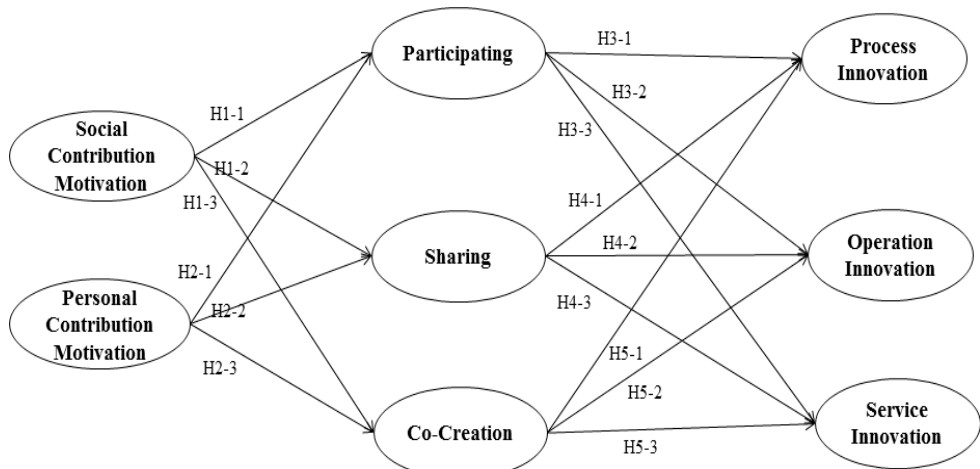

**Figure 1.** Suggested research model.

**4. Method**

*4.1. Sampling and Data Collection*

A pretest was administered to 30 corporate employees who used CI in Korea. The measurement tools were based on the literature review related to participants' motivations and were also measured as exogenous variables; CI, acceptance intention, and incremental innovation were also measured as endogenous variables. The study used the convenience sampling of corporate employees who use CI from about 2000 corporate panel members. Then, they were contacted by e-mail and telephone and asked to participate in the study. A description of CI was placed in the first section of the questionnaire. A total of 1500 corporate employees participated in this study and were given gift cards by the investigator.

As seen in Table 1 below, over 75% of participants were 20–49 years of age and over 90% had received college-level instruction. On average, respondents occupied positions in accounting and management, strategic planning, production and research, and operations.

**Table 1.** Demographic profiles.

| *n* = 1500 | | Frequency | % |
|---|---|---|---|
| Sex | Male | 760 | 50.7 |
| | Female | 740 | 49.3 |
| Education | High School | 102 | 6.8 |
| | College Student | 367 | 24.5 |
| | Bachelor's degree | 678 | 45.2 |
| | Over M.A Degree | 353 | 23.5 |
| Ages | 20–29 years | 361 | 24.1 |
| | 30–39 years | 401 | 26.7 |
| | 40–49 years | 367 | 24.5 |
| | 50–59 years | 321 | 21.4 |
| | Over 60 years | 50 | 3.3 |
| Position | Accounting and Management | 455 | 30.3 |
| | Operations | 242 | 16.1 |
| | Strategic Planning | 318 | 21.2 |
| | Production and Research | 269 | 17.9 |
| | Others | 216 | 14.4 |
| Firm Size | 10 and fewer employees | 219 | 14.6 |
| | 10–30 employees | 284 | 18.9 |
| | 30–50 employees | 138 | 9.2 |
| | 50–100 employees | 199 | 13.3 |
| | 100–300 employees | 272 | 18.1 |
| | Over 300 employees | 389 | 25.9 |

*4.2. Instrument Construction*

4.2.1. Exogenous Variables

Contribution motivations were measured by social and personal contribution motivations as exogenous variables. Social contribution motivations can be defined as embracing the objective of collective intelligence to share information, edit or modify the content, resolve curiosity, co-create information, and acquire fame. Questions about social contribution motivations used in this study consist of a total of five items developed in light of the existing literature [25,26]. Personal contribution motivations comprise a total of five items developed from existing research by defining the degree of desire to acquire new information, expectation about another's responses, compensation, showing the ability to offer information, and work efficiency [20,21,23,24].

4.2.2. Endogenous Variables: Collective Intelligence, Incremental Innovation Capability

Collective intelligence was classified with collective intelligence building through participation, sharing, and co-creation. Collective intelligence building through participation included users' positive participation in space to develop personal knowledge, their career and recognition of their capability. Collective intelligence building through sharing included tools to seek solutions to their problems and share and acquire their new knowledge or information. Collective intelligence building through co-creation included users' operation or management forming knowledge, or information through collective intelligence. Collective intelligence is assumed to be a collection of rational judgments made by individuals and is considered to be a source of better judgment than is possible with a small group of specialized professionals or the combined capabilities of a single individual. Measurement tools for collective intelligence used in this study consist of a total of 13 items taken from the existing literature [15,18,24,39].

Scale items used in the study were adapted from previous studies. Twelve items for incremental innovation were developed or adopted from previous studies [8,32–34]. Incremental innovation was

measured by work process innovation, operation innovation, and service innovation. Work process innovation included the degree of improvement over the past, management process, work efficiency and cost reduction. Operation innovation represented the degree of improvement of welfare, work satisfaction, and relationships with others in the organization. Service innovation was defined as the degree of improvement in customer service, response to complaints, and satisfaction. All items used in this study were scored on 5-point Likert scales.

## 5. Data Analysis

### 5.1. Assessment of the Measurement Model

To analyze the data, the study used EQS6b and SPSS. In order to verify the hypotheses proposed in this study, correlations, validity and reliability were examined. We conducted verification tests to determine tests for the measurement model's validity using EQS6b. As shown in Tables 3 and 4, the standard acceptance norm satisfies the requirement of reliability and validity by suggested by Bagozzi and Yi [43] and Hair et al. [44] and composite construct reliability and average variance extracted (AVE) following Fornell and Larker [45]. Discriminant validity was assessed by comparing the correlation of components to AVE. As seen in Table 2, the result of Bartlett's test of sphericity was found to be significant ($\chi^2$ = 7959.3, $df$ = 741, $p < 0.001$), while the Kaiser–Meyer–Olkin measure of sampling adequacy was 0.947 for the variables [44]. The results of the EFA (exploratory factor analysis) are described in Table 3. An exploratory factor analysis of all of our scale items revealed two factors explaining 56.08% of the variance in our study's constructs, with the first factor explaining 30.42% and the last factor explaining 25.67% of the total variance for independent variables. Discriminant validity was assessed by comparing the correlation of components to AVE.

**Table 2.** Statistics of construct items.

| Construct | Survey Measures |
|---|---|
| Social contribution motivation | I wish to contribute my knowledge for the purpose of sharing it with others (sharing contribution) |
| | I wish to add or correct wrong information in the public based on my knowledge (addition · correction) |
| | I wish to help others find answers to their questions (answer questions) |
| | I wish to exhibit my knowledge to large numbers of people (knowledge exhibition) |
| | I like to cooperate with others to create knowledge (knowledge collaboration) |
| Personal contribution motivation | I can acquire new knowledge and skills by contributing (skill acquisition) |
| | I find it helps my career (career development) |
| | I am curious about the responses of others regarding my knowledge (response of others) |
| | I like to be compensated for the provision of knowledge (tangible/intangible compensation) |
| | I wish to be recognized for my capability and seek a greater reputation (recognition and reputation) |

**Table 2.** *Cont.*

| Construct | Survey Measures |
|---|---|
| Participation | I can easily express my work-related opinions using collective intelligence |
| | I can easily post work-related opinions using collective intelligence |
| | I can easily solicit work-related opinions from other organization members using collective intelligence |
| | I can express work-related ideas using collective intelligence. |
| | I can post unique work-related opinions to organization members using collective intelligence |
| Sharing | I use collective intelligence to acquire new knowledge or information |
| | I use collective intelligence to share knowledge or information with other people |
| | I use collective intelligence to acquire knowledge or information not found in other places |
| | I use collective intelligence to find solutions to my problems |
| Co-Creation | I like to post new writings, pictures, and videos using collective intelligence |
| | I use knowledge or information from collective intelligence after correcting, editing, or reprocessing it |
| | I directly operate or manage a means of collective intelligence |
| | I make an effort to continually post new information to collective intelligence |
| Process Innovation | In-house rules and work processes have improved compared with the past |
| | The management process has improved compared with the past. |
| | The purposes of reducing costs have improved compared with the past. |
| | The work efficiency has improved compared with the past. |
| Operation Innovation | The welfare of employees has improved compared with the past. |
| | Work satisfaction among members have enhanced compared with the past. |
| | The work of members has improved compared with the past. |
| | Personal relationships among workers have improved compared with the past. |
| Service Innovation | Visiting customers have increased compared with the past |
| | Customer satisfaction raised compared with the past. |
| | Our response to customer complaints have improved compared with the past. |
| | The speed of customer service has improved compared with the past. |

**Table 3.** Results of factor analysis.

| Independent Variables | | | Dependent Variable | | |
|---|---|---|---|---|---|
| **Construct** | **Items** | **F.L** | **Construct** | **Items** | **F.L** |
| Social Contribution Motivation | SCM1 | 0.758 | Co-Creation | COC1 | 0.845 |
| | SCM2 | 0.778 | | COC1 | 0.863 |
| | SCM3 | 0.764 | | COC1 | 0.846 |
| | SCM4 | 0.791 | | COC1 | 0.916 |
| | SCM5 | 0.794 | Dependent Variables | | |
| | | | Process Innovation | PI1 | 0.938 |
| Personal Contribution Motivation | PCM1 | 0.793 | | PI1 | 0.934 |
| | PCM2 | 0.802 | | PI1 | 0.837 |
| | PCM3 | 0.812 | | PI1 | 0.747 |
| | PCM4 | 0.789 | | PI1 | 0.840 |
| | PCM5 | 0.762 | Operation Innovation | OI1 | 0.847 |
| Mediated Variables | | | | OI2 | 0.830 |
| Participation | PAR1 | 0.734 | | OI3 | |
| | PAR2 | 0.690 | | OI4 | 0.808 |
| | PAR3 | 0.725 | Service Innovation | SI1 | |
| | PAR4 | 0.702 | | SI2 | |
| | PAR5 | 0.723 | | SI3 | |
| Sharing | SHA1 | 0.693 | | SI4 | |
| | SHA2 | 0.723 | | | |
| | SHA3 | 0.704 | | | |
| | SHA4 | 0.791 | | | |

| Independent Variables | | | Dependent Variables | | |
|---|---|---|---|---|---|
| Factor | Eigenvalues | % of Variance | Factor | Eigenvalues | % of Variance |
| Factor 1 | 4.183 | 30.416 | Factor 1 | 4.355 | 27.217 |
| Factor 2 | 1.425 | 25.663 | Factor 2 | 3.849 | 24.056 |
| % of total variance extracted | 56.079 | | Factor 3 | 3.064 | 19.149 |
| Mediated Variables | | | % of total variance extracted | 70.422 | |
| Factor | Eigenvalues | % of Variance | | | |
| Factor 1 | 3.645 | 29.042 | | | |
| Factor 2 | 2.929 | 22.528 | | | |
| Factor 3 | 1.761 | 13.546 | | | |
| % of total variance extracted | 64.116 | | | | |

Note: F.L: Factor Loadings, SCM: social contribution motivation, COC: Co-creation, PCML: personal contribution motivation, SHA: sharing, PAR: participation, PI: process innovation, OI: operation innovation, SI: service innovation.

As seen in Table 4, the Cronbach's alpha mean for all concepts is above 0.7 [46]. The study's AVE also satisfies the standard of 0.5 for the requirement for convergent validity.

**Table 4.** Internal consistency of the constructs.

| Variables | Items | M | S.D | $\alpha$ | C.R |
|---|---|---|---|---|---|
| Social Contribution | 5 | 3.71 | 0.55 | 0.804 | 0.882 |
| Personal Contribution | 5 | 3.68 | 0.54 | 0.835 | 0.930 |
| Participating | 5 | 3.69 | 0.52 | 0.803 | 0.880 |
| Sharing | 4 | 3.74 | 0.56 | 0.810 | 0.885 |
| Co-Creation | 4 | 3.16 | 0.81 | 0.844 | 0.898 |
| Process Innovation | 4 | 3.44 | 0.67 | 0.837 | 0.937 |
| Operation Innovation | 4 | 3.40 | 0.74 | 0.893 | 0.958 |
| Service Innovation | 4 | 3.38 | 0.78 | 0.916 | 0.967 |

Note: M: Mean; S.D; Standard deviation; $\alpha$: Cronbach's alpha; C.R: Composite reliability.

As seen in Table 5, the extracted AVE is between 0.648 and 0.764, and the means of the squares of the correlation coefficients are between −0.001 and 0.482, which results in an AVE that is higher than the means of the squares of the correlation coefficients ($r^2$), also ensuring that the data collected for verification have sufficient discriminant validity [43].

**Table 5.** Analysis of discriminant validity using average variance extracted.

| | AVE | 1 | 2 | 3 | 4 | 5 | 6 | 7 | 8 |
|---|---|---|---|---|---|---|---|---|---|
| 1 | 0.673 | 1 | | | | | | | |
| 2 | 0.764 | 0.013 | 1 | | | | | | |
| 3 | 0.769 | 0.482 | 0.111 | 1 | | | | | |
| 4 | 0.648 | 0.328 | 0.063 | 0.366 | 1 | | | | |
| 5 | 0.687 | 0.324 | 0.139 | 0.041 | 0.012 | 1 | | | |
| 6 | 0.749 | 0.001 | 0.200 | 0.041 | 0.011 | 0.120 | 1 | | |
| 7 | 0.751 | 0.159 | 0.278 | 0.119 | 0.014 | 0.133 | 0.387 | 1 | |
| 8 | 0.760 | 0.005 | 0.107 | 0.022 | 0.060 | 0.010 | 0.344 | 0.507 | 1 |

Note: All coefficients: Squared the correlation coefficients, 1: Social Contribution, 2: Personal Contribution, 3: Participating, 4: Sharing, 5: Co-Creation, 6: Process Innovation, 7: Operation Innovation, 8: Service Innovation. AVE, average variance extracted.

## 5.2. Tests of Hypotheses

As proven previously, hypotheses for this study based on the research model satisfy the advised base values. The goodness of fit of the model hypotheses yielded $\chi^2 = (11,107.2) = 652$, $p = 0.000$, CFI = 0.946, NFI = 0.908, NNFI = 0.946, GFI = 0.878, AGFI = 0.858, SRMR = 0.110, RMSEA = 0.048, which means that the model's goodness of fit satisfies the advised base values.

To test structural relationships, the hypothesized causal paths were estimated. Eight hypotheses were supported. The results are shown in Figure 2 and Table 6. For H1-1, H1-2, and H1-3, the results indicate that social contribution motivations are closely related to collective intelligence building through participation. The suggested path was statistically significant in the hypothesized direction (social contribution motivations, with a standardized path coefficient for collective intelligence building through participating: $\gamma = 0.213$, $p < 0.001$ for H1-1). Thus, hypothesis H1-1 was supported. Social contribution motivations are closely related to collective intelligence building through sharing. The suggested path was significant in the hypothesized direction (social contribution motivations, with a standardized path coefficient for collective intelligence building through sharing: $\gamma = 0.317$, $p < 0.001$ for H1-2). Thus, hypothesis H1-2 was supported. Social contribution motivations are closely related to collective intelligence building through co-creation. The suggested path was statistically significant in the hypothesized direction (social contribution motivations, with a standardized path

coefficient for collective intelligence building through co-creation: $\gamma = 0.325$, $p < 0.001$ for H1-3). Thus, hypothesis H1-3 was supported.

For H2-1, H2-2, and H2-3, the results indicate that personal contribution motivations are closely related to collective intelligence building through participating. The suggested path was statistically significant in the hypothesized direction (personal contribution motivations, with a standardized path coefficient for collective intelligence building through participation: $\gamma = 0.492$, $p < 0.001$ for H2-1). Thus, hypothesis H2-1 was supported. Personal contribution motivations are closely related to collective intelligence building through sharing. The suggested path was statistically significant in the hypothesized direction (personal contribution motivations, with a standardized path coefficient for collective intelligence building through sharing: $\gamma = 0.664$, $p < 0.001$ for H2-2). Thus, hypothesis H2-2 was supported. Social contribution motivations are positively related to collective intelligence building through co-creation. The suggested path was significant in the hypothesized direction (personal contribution motivations, with a standardized path coefficient for collective intelligence building through co-creation: $\gamma = 0.532$, $p < 0.001$ for H2-3). Thus, hypothesis H2-3 was supported.

**Table 6.** Summary of hypothesis tests.

| Hypothesis | S.E | Standardized Coefficient | Support |
|---|---|---|---|
| H1-1: Social Contribution -> Participating | 0.035 | 0.213 *** (0.278)/z = 7.763 | Yes |
| H1-2: Social Contribution -> Sharing | 0.144 | 0.317 *** (0.402)/z = 10.211 | Yes |
| H1-3: Social Contribution -> Co-Creation | 0.052 | 0.325 *** (0.503)/z = 11.012 | Yes |
| H2-1: Personal Contribution -> Participating | 0.054 | 0.492 *** (0.571)/z = 12.352 | Yes |
| H2-2: Personal Contribution -> Sharing | 0.071 | 0.664 *** (0.744)/z = 13.423 | Yes |
| H2-3: Personal Contribution -> Co-Creation | 0.056 | 0.532 *** (0.602)/z = 12.985 | Yes |
| H3-1: Participating -> Process | 0.055 | 0.249 *** (0.307)/z = 8.974 | Yes |
| H3-2: Participating -> Operation | 0.063 | 0.210 *** (0.346)/z = 13.321 | Yes |
| H3-3: Participating -> Service | 0.058 | 0.356 *** (0.472)/z = 15.742 | Yes |
| H4-1: Sharing -> Process | 0.052 | 0.291 *** (0.395)/z = 14.213 | Yes |
| H4-2: Sharing -> Operation | 0.048 | 0.259 *** (0.325)/z = 10.855 | Yes |
| H4-3: Sharing -> Service | 0.055 | 0.309 *** (0.412)/z = 10.512 | Yes |
| H5-1: Co-creation -> Process | 0.056 | 0.307 *** (0.410)/z = 10.103 | Yes |
| H5-2: Co-creation -> Operation | 0.068 | 0.306 *** (0.406)/z = 11.317 | Yes |
| H5-3: Co-creation -> Service | 0.075 | 0.392 *** (0.502)/z = 12.308 | Yes |

Notes: *** $p < 0.05$, (unstandardized) coefficient.

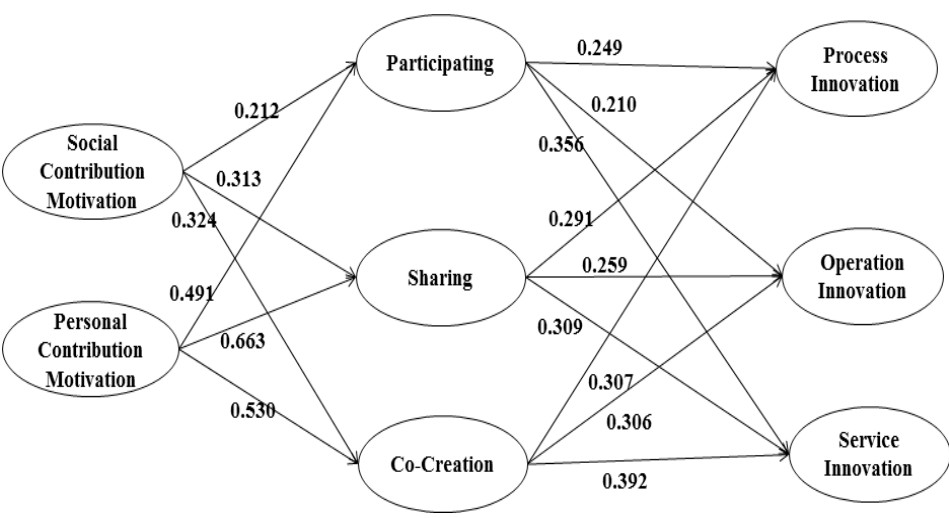

**Figure 2.** Results of suggested research model with path coefficients.

For H3-1, H3-2, and H3-3, the results indicate that collective intelligence building through participation is closely related to process, operation, and service innovation as sub-factors of incremental innovation. The proposed path was statistically significant in the hypothesized direction (collective intelligence building through participation, with a standardized path coefficient for process, operation, and service innovation: $\gamma = 0.249$, 0.210, 356, $p < 0.001$ for H3-1, H3-2, and H3-3, respectively). Thus, hypotheses H3-1, H3-2, and H3-3 were supported.

For H4-1, H4-2, and H4-3, the results indicate that collective intelligence building through sharing is closely related to process, operation, and service innovation as sub-factors of incremental innovation. The suggested path was statistically significant in the hypothesized direction (collective intelligence building through sharing, with a standardized path coefficient for process, operation, and service innovation: $\gamma = 0.291$, 0.259, 309, $p < 0.001$ for H4-1, H4-2, and H4-3, respectively). Thus, hypotheses H4-1, H4-2, and H4-3 were supported.

For H4-1, H4-2, and H4-3, the results indicate that collective intelligence building through co-creation is closely related to process, operation, and service innovation as sub-factors of incremental innovation. The suggested path was statistically significant in the hypothesized direction (collective intelligence building through co-creation, with a standardized path coefficient for process, operation, and service innovation: $\gamma = 0.307$, 0.306, 392, $p < 0.001$ for H5-1, H5-2, and H5-3, respectively). Thus, hypotheses H5-1, H5-2, and H5-3 were supported.

## 6. Discussion

This study proposes an acceptance model for corporate CI. The motivators of CI associated with corporate employees were divided into social and individual contribution motivations. We found that the motivation for using CI was more likely to come from social contribution motivations than from individual contribution motivations. CI was divided into CI-building through participation, sharing, and co-collaboration, and its relationship to incremental innovation was studied. Additionally, corporate employees were found to have the desire to use CI to foster innovation in their work processes, operations, and services. The results confirmed that CI is derived from the activities of internal and external participants who cooperate and compete in the process of finding solutions to difficult problems that typically cannot be solved by a specialized pool of professionals within a corporate organization. The study suggests that CI is one of the critical tools for developing work processes, operation, and service sector in organizations. The study found that social and personal contribution motivations are positively related to CI-building through participation, sharing, and co-creation. People who like to share knowledge with others and acquire new knowledge have the tendency to participate, share, and manage knowledge and information with others using CI. CI- building, through participation, sharing, and co-creation, is positively related to process, operation, and service. An organization with a strong tendency to participate, share, and operate a means of CI has a higher possibility of developing process, operation, and service innovation outcomes. CI was found to affect work process, operations, and service innovation. This suggests that CI is pursued more actively as work processes are made more innovative, and the improvements are greater in the performance of work processes, work procedures, work efficiency, customer satisfaction, and services. CI is built on some combinations of individual members' attributes and group structures, processes, and norms. It plays an important role in improving decision-making in groups; it also produces higher group performance and predicts a group's future performance on more complex tasks and problems [35,40]. It was also found to affect work processes, operation, and service innovations.

### 6.1. Theoretical Implications

The study attempted to identify cause-and-effect relationships associated with CI and explain a theoretical model that covers the relationship between CI and incremental innovation. The casual relationship and theoretical model suggested in this study might be valuable assets for further studies exploring the classification of CI. They might also help in defining determinant factors leading to CI

and in understanding how these factors relate to each other and to innovation capabilities in enterprises. The results of this study provide valuable information for effective decision-making based on CI, as well as understanding the cause-and-effect relationships associated with CI in the academic area.

Regarding the result indicating the association of CI with incremental innovation, if corporations pursue active participation, sharing, and collaboration for using CI services, then from an organizational or corporate management standpoint, the effects of decision-making processes, as well as product and service development efforts will improve. The study indicates that CI involves a process in which an individual's idea is combined with and evolves with other ideas to produce a force greater than what can be expected from simply, including the intellectual capabilities of each individual. In other words, a synergy effect is produced. When corporate employees work in an environment where CI is highly developed, work procedures or efficiency may differ depending on the onset of CI. This raises the importance of CI within an organization and implies the importance of finding means to vitalize CI. In a digital society, a corporate organization improves performance, competitiveness, and productivity through the division of knowledge. Corporations are formed from organic connections between groups (organizations). In light of these considerations, we hope that this study's results will help corporations tap into CI as a resource, improve functioning, and develop its application in the corporate context. The more the knowledge that was shared with colleagues or strangers, the more the tasks that will be undertaken via co-collaborative or creative efforts with colleagues. This result indicates that personal contribution motivation of CI (e.g., career development) has a higher perception than social contribution motivation, including knowledge sharing or collaboration with others. Such corporations vitalize their services by raising the quality of information and knowledge shared in their communities. Contribution motivations that take the characteristics of knowledge and information contributors into consideration also need to be developed.

## 6.2. Managerial Implications

From a corporate standpoint, continual and systematic efforts are required for establishing an active online community that creates CI and for improving levels of participation, openness, sharing, and collaboration. More references or external links that support the contribution by knowledge or information providers must be created. In addition, this study found that the pursuit of active participation by knowledge contributors and encouragement of greater collaboration implicitly contribute to the improvement in the quality and credibility of knowledge or information. For the corporate employees to develop products or ideas, the belief that new knowledge can be created during the process of fine-tuning of various knowledge bases must be strengthened while pursuing the creation of new knowledge itself. This seems to point to a need for developing CI services that will raise levels of participation, sharing, and collaboration while also inspiring its contribution motivations.

The rise in success cases involving CI has resulted in the expansion of its application across all corporate activities, including product development, marketing, product manufacturing, and customer support. Corporations willing to introduce CI must establish a platform from which it can work and establish a network that must be continually managed during the process. The establishment of windows of communication, an open and cooperative corporate culture, and participation by organizational members with varying knowledge and experience are needed, where incentives for knowledge or information sharers may also prove to be useful.

The CI within cyberspace, which is the topic of interest in this study, has been the focus in various academic fields. Human societies have cultivated CI through the utilization of science and technology, while communicating shared intellectual capabilities and assets with one another. Through CI, humanity can overcome space and time limitations and achieve true integration that will lead to a new level of evolutionary accomplishment. Therefore, exploring CI in cyberspace and its causal relationship with any other outcome is a subject for further study. CI is difficult to achieve in traditional hierarchical structures. It requires freedom, volunteerism, and candid opinion formation. Collaborative leadership

is important in such a context. Leadership that lends an ear to even obnoxious or seemingly nonsensical thoughts is needed.

*6.3. Future Direction and Limitations*

As an example, Procter and Gamble utilizes CI to reduce its share of research and development costs, while continuing with good performance. On the other hand, Monsanto spends a substantial amount of money on research and development without producing any major products and faces a crisis. Smooth mutual engagement, feedback, and active participation are methods that can raise CI. Additionally, corporate cultures must find efficient spaces in which employees can communicate.

This study was subject to a significant limitation. It examined two different motivations for knowledge development: CI and incremental innovation. Thus, the causal relationships considered in this study might be weak. Future research on similar topics can consider individual goals or organizational goals as their objectives. Regarding the sample used in this study, the effect of CI might not be representative, because incremental innovation was drawn from only one outcome. The current limitations of the study are weak, and issues with business performance, etc., are not addressed. Therefore, additional studies with appropriate controls are needed to apply these results to theoretical models of online behavior.

Another limitation is that the measurement tools to measure CI provided in this sample are not strong. The items to measure CI might cause certain problems when clarifying CI from individual and organizational standpoints. Thus, CI should be applied to similar topics to increase its statistical validity, such as content or face validities. The study did not address CI issues for different types of organizations. The relationship between the effectiveness of CI and its outcomes, in providing their size (e.g., number, industries they operate, technologies they use for CI, strategy with respect to idea generation, growth over the last years) should be developed. Finally, the cause-and-effect relationships are not considered clearly and strongly, as the present study is cross-sectional and non-experimental.

**Author Contributions:** C.-H.J. wrote the paper and worked with J.-Y.L. to conceive and design the experiments J.-Y.L. and C.-H.J. performed the experiments and analyzed the data; J.-Y.L. and C.-H.J. contributed to parts of the experiments and the conclusions. Both authors made contributions to the work in this study.

**Funding:** This research received no external funding.

**Conflicts of Interest:** The authors declare no conflict of interest.

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
