# Peer review of "How Collective Intelligence Fosters Incremental Innovation"

_2199-8531, doi:10.3390/joitmc5030053_

Round 1

Reviewer 1 Report

Thank you very much for providing me with this opportunity to review the draft. Overall discussion is quite interesting to me and I enjoyed a lot. However, I am afraid I found several points to be improved before publication, especially the flow of the authors' discussion is not strong enough to be persuasive. Moreover, the critical careless mistakes were also found. Please refer to my comments below.

2: Incremental innovation capability in the title does not fit with the actual scope of the analysis. The authors analyzed incremental innovation output.

40: The authors should not mention cause-and-efect relationships this much clearly and strongly, as the present study is cross-sectional and non-experimental. At least, this limitation should be mentioned at the end of the draft.

41-44: Explore the antecedent of collective intelligence is out of scope of a theoretical model that explains the relationship between collective intelligence and incremental innovation.

46: Rsult should be result. Marking should be making.

60 and others: Mixed uses of collective intelligence and CI.

95: It was easier to understand the flow of discussion if these three types were introduced much earlier in this section, for example, just after the definition, especially because the three types were argued rather in a fragmented way in the previous part of the section. But still one question is how the three types are linked to three sub-factors of the construct for the empirical analysis later in the draft, though the author made some statements on the three sub-factors. From the author's perspective, that may be obvious, but not clear to me and I guess, many of the other first time readers.

104-106: In line with the comment just above, is participation (one of the sub-factors) related to all the three types or to the co-creating collective intelligence? That is unclear to me and make me feel confused.

112-146: This sub-section (2.2) again is not good to link the overall theoretical discussion and sub-factors of the motivation construct. My candid impression is not showing the organized discussion toward the justification of classifying sub-factors, but shifting one perspective to another simply and not effective to justify the sub-factors. The authors may have their own logic, but at least not clear to me.

147-168: There is no serious problem about this sub-section in terms of the flow of the discussion, which is different from the previous two sub-sections.

156: It is much better to start a new paragraph after [8,32]. and combine it with the next paragraph.

174: Confusing is that, although knowledge sharing in this sub-section covers all the three sub-factors of collective intelligence, the authors did not provide understandable explanation on that point. It is necessary that they are using 'sharing' in narrow and broad sense. The former matches with the scope of the sub-factors of 'sharing' while the latter covers participation, co-creation as well as sharing.

192: This first sentence is rather misleading. It is likely to suggest the existence of moderators that may affect the effectiveness of collective intelligence. This argument is against your hypotheses (H3, H4 and H5). More in a straight forward way, collective intelligence is sometimes positively related to incremental innovation but in other cases no positive relation can be found. If the authors justify that the current sample satisfy the condition(s) leading to the positive and significant relation, the sentence should be maintained but I could not find the relevant argument.

216-220: It may be acceptable to hypothesize specifically like this based on the discussion only on the general relationship between collective intelligence and incremental innovation.  But at least they should clearly explain this general discussion can be applicable to the specific ones, too.

239: Very basic information on the sample is missing. Are the participants from a single country or multiple number of countries? If the former is the case, what country?

245: How many were contacted in order to collect the data from 1,500 employees?

274: The items seem to measure innovation totally rather than only incremental innovation, although substantially this point may not be serious, as we may have much more cases of incremental rather than radical innovation. But some elaboration is required.

281: Must be Table 2. Accordingly the following table numbers should be changed. Moreover, it is difficult to find which measures are under which constructs. Not only this table, other tables should be revised in order to make readers understand what you want to inform clearly.

293-295: Why did the authors show the EFA result of independent variables only, not showing those of two other sets of variables?

318: 11107.2 cannot be equal to 652. The latter must be degree of freedom. Describe that way accordingly.

328: More than eight hypotheses were supported and no hypothesis was not supported. This is too much serious beyond the careless mistake.

374-376: The sentence is contradictory to the results and the sentence in lines 379-381.

392: as mentioned above, some statement on the limitation of the research design should be included.

394: 'Might provide', not 'might be provide'.

391-436: The discussion is too much beyond the results of the present study.The authors may include something beyond for showing future research directions, but the main issue for implications should be directly related to the results.Usually the future research directions are argued in the separate section.

436-481: Again, not relevant to the specific study results. Moreover, even though the implications should focus on the effectiveness of improving motivation based on the framework and the results, the arguments are more likely to start from collective intelligence.   

Author Response

Issue Number: JOItmC-553358

We thank the reviewers for their insightful comments in improving this paper and further highlighting the importance of this avenue of research. Each point made by each reviewer has been addressed and the response is outlined below. The revised part is highlighted in red color.

Major comments and suggestions:

In comment 1, both reviewers advise the author to hire a professional editing service to enhance the readability and logicality of sentences and paragraphs and to correct typographical errors.

Response: We hired a professional editing service provider to address grammatical, typographical, and other errors, and enhance the readability, logicality, and flow of sentences and paragraphs.

In comment 2, one reviewer asks the author to revise the title to match the study’s main framework.

Response: In response to this comment, the authors have revised the title as follows: “How Collective Intelligence Fosters Incremental Innovation Output”

In comment 3, one reviewer asks the author to logically revise the main purpose of the study in introduction section.

Response: In response to this comment, the authors have enriched the main purpose of the study on page 1.

In comment 4, both reviewers ask the author to revise the literature review to develop the concepts of CI, particularly those that propose definitions of the construct.

Response: In response to this comment, the authors have revised the literature review and enhanced the argumentation related to the concepts and terms associated with CI and other constructs used in the study on pages 1 and 3.

In comment 5, reviewer asks the author to develop the hypothesis session.

Response: In response to this comment, the authors have developed and enriched the hypotheses on pages 3–4 based on the relevant articles and comments. They have also revised hypotheses 3–5

In comment 6, reviewer asks the author to revise the method issues such as sampling, questionnaires, and number Table.

Response: In response to this comment, the authors have revised the Method section, focusing on the sampling method, instrument, and EFA results of all variables described in Table 3, and have fixed table numbers from pages 3–8.

In comment 7, both reviewers ask the author to revise the discussion, implication by improving and reorganizing the presentation of the study’s implications. Also adding limitation and future direction section.

Response: In response to this comment, the authors have revised and enriched the discussion and the presentation of the implications by dividing the Discussion section into Theoretical Implications, Practical Implications, and Limitations and Directions for future research from pages 8–13, focusing on managerial implications based on the results of the relationship between CI and incremental innovation. Particularly, the cause-and-effect relationships used in the study are described in the Limitation section for further study.

Thank you for your wonderful comments, which guided our paper’s revision.

Reviewer 2 Report

Dear Authors,

Thank you for the opportunity to review your article. This is an interesting paper that addresses an important topic for the companies, which should reinvent themselves to be innovative and adaptable to contemporary environmental changes.

The article provides interesting insights into the empirical examining of the relationship between Collective Intelligence and Incremental Innovation, in order to understand how collective intelligence is related to incremental innovation, such as work processes, operations, and services innovation.

The paper is generally well-written and therefore easy to understand and follow.

The Literature review was clearly presented. Authors precisely justified the topics and their importance, by referring to appropriate theoretical contextualization of the issues.  

It should be noted that the hypotheses are rigorous presented, however, I would like to suggest the following assumption that can be take into consideration.

Beside the fact that collective intelligence play an important role in improving corporate work processes, management procedures, and customer satisfaction it is expected that collective intelligence-building leads to the information entropy process at the organizational level.The relationship between collective intelligence and incremental innovation may produce different forms of disorders in organizational system, based on the uncertainty of information. Thus an entropy process is emerged. Under these conditions, the collective intelligence-building requires internal control elements to ensure that the organizational system evolves towards higher-order levels, in terms of the amount of information.

To perform a more comprehensive approach it should be useful to consider the entropy phenomenon related to work process, operation, and service innovations. Thus, by controlling entropy process inside certain limits is expected to improve the collective intelligence and its relationship with incremental innovation. 

I also appreciate that the research methods are adequate and well justified.

In the Discussion section the result of the study indicated theoretical and managerial implications that provide a valuable foundation to implement the research model suggested by the authors.

However, I suggest introducing a brief Conclusions section in order to summarize the most important arguments and facts provided by the research.

I think the paper has an interesting focus and provide an original approach to cover the lack of research on identifying cause-and-effect relationships associated with collective intelligence.

Best of luck with your manuscript.

Regards,

Author Response

(The authors gave the same response as above.)

Round 2

Reviewer 1 Report

I appreciate the authors' efforts for thorough revisions. Let me make some more comments as follows. I am so sorry. I should have read the draft more carefully and noticed those issues at the first round.

2: I am so sorry that I did not notice, but it seems to be problematic that the current article title does not contain one of the major variables in the analysis, contribution motivation. Readers expect the framework without contribution motivation and no hypothesis related to it. Honestly speaking, I have no clear and good idea but a conventional way of title telling the major variables one by one, such as 'Contribution Motivation, Collective Intelligence and Incremental Innovation (Output)', though there may be a better one. (I think you should avoid the term, incremental innovation capability but not necessary to mention 'output', rather more generally 'incremental innovation', as for the two other variables, you did not mention that much in detail.

7-8: I am sorry again, but I am afraid I have just noticed contribution motivation must be mentioned in this sentence. It is not acceptable to mention it only in the first and the last sentence, especially because the authors developed some hypotheses regarding this factor.

8-10: The cause-effect relationship (though we should not argue confidently due to the inappropriate research design) is not precise. Based on the plausible process, you should revise like this way.

'This suggests that, if collective intelligence is pursued more actively, work processes, operations and services are made more innovative'.

The original version seems to suggest the reverse causality.

Moreover, regarding the following part, 'the improvements in the performance of work processes, work procedures, work efficiency, customer satisfaction, and services are greater': this part is obviously the consequences of incremental innovation and so beyond the scope of the present study. that is why inappropriate to mention in the abstract.

13-14: Does not make sense to me. Do you mean 'management is required to develop further contribution motivation as an important aspect of the characteristics of knowledge and information contributors'.

37: 'The relationships between CI and incremental innovation were examined.': not necessary as it is without contribution motivation while just repeating the CI-incremental innovation relation.

82-83: Overall descriptions in this paragraph are good. But please be careful! You must make clear somewhere in the paragraph that contributing CI is as same as participation CI, even if that may be obvious to the authors.

97-98: You don't need to insist on IT here. The overall discussion is more general, not specific about the IT aspect.

197-198: Usually we should not include the phrase such as 'according to one reviewer'. Rather refer to the published academic work(s).

350-351: According to the path coefficients, CI was more likely to come from individual contribution motivations than from social contribution motivations, opposite from what is mentioned in the draft. Moreover, the term 'motivation for using CI' itself and the idea of its relationship with the two types of motivations are logically very much confusing.

354: Too bold to tell 'online communities used by corporations are products of CI' based on the data for the present study. I am afraid no relevant information was obtained for this empirical study. If you want to argue like this, the additional supporting evidence is necessary.

355: Whether corporate employees have the desire to use CI cannot be discussed by the main result on the relationship between contribution motivation and CI. You may argue if the level of such motivation is high but we cannot be sure if the means (3.71 and 3.68) are undoubtedly high. Anyway this point is not the major issue for discussion, though it may be worth mentioning somewhere in the discussion section.

362-363: Frankly speaking, I cannot really understand why the authors don't feel uncomfortable when using the term, knowledge sharing in both broad and narrow senses without the clear distinction. It is too much confusing and logically unacceptable. Really hope that the authors understand the seriousness of this problem.

366-369: Same as the comments on the abstract.

374-375: Beyond the scope of the present study framework.

6.1 'Theoretical Implications' did not contain the substantial theoretical implications derived from the results. For 6.2 as well, all the implications should start from how to enhance contribution motivation rather than from CI, along with the framework of the present study. But this basic rule was not followed and then the discussion was confusing. If you want to focus on the effect of CI on incremental innovation only, from the beginning, you did not have to analyze the motivation. Many of the descriptions are not for the implications, more appropriate as the background information to be explained in the introduction section.

443-447: Cannot understand the additional value of the paragraph, particularly for this subsection.

448-454: Cannot follow the flow of the discussion. The scopes seem to change illogically and feel at a loss.

Need the proofreading by the better professional English editor or the same editor but with much more attentions to the logical discussion (usually the higher rate is likely to be required). Otherwise,

Author Response

Issue Number: JOItmC-553358

We thank the reviewers for their insightful comments in improving this paper and further highlighting the importance of this avenue of research.

Major comments and suggestions:

In comment 1, reviewer advises the author to hire a professional editing service to enhance the readability and logicality of sentences again.Response: We hired a professional editing service provider to enhance the readability, logicality, and flow of sentences and paragraphs again. In comment 2, one reviewer asks the author to change the title to match the study’s main framework.  Response: In response to this comment, the authors have changed the title as follows: “How Collective Intelligence Fosters Incremental Innovation” In comment 3, one reviewer asks the author to develop the misleading or illogical redundancy sentence in all section Response: In response to this comment, the authors have enriched or removed the sentences which are not match for the text. In comment 4, reviewer ask the author to revise the implication by improving and reorganizing the presentation of the study’s implications.   Thank you for your wonderful comments, which guided our paper’s revision. Response: In response to this comment, the authors have revised the presentation of the implications. Author have to enrich the part in summarizing the section. Also the author removed unnecessary sentences at the section.

This manuscript is a resubmission of an earlier submission. The following is a list of the peer review reports and author responses from that submission.